# Gynae-Oncology Surgeons’ Preparedness to Undertake Colorectal Procedures during Cytoreductive Surgery for Ovarian Cancer: A Cross Sectional Survey

**DOI:** 10.3390/jcm11216233

**Published:** 2022-10-22

**Authors:** Daniel Huddart, Savithri Rajkumar, Desiree Kolomainen, Gautam Mehra, Rahul Nath, Ahmad Sayasneh

**Affiliations:** 1Faculty of Life Sciences & Medicine at Guy’s, The School of Life Course Sciences, King’s College London, London WC2R 2LS, UK; 2Department of Gynaecological Oncology, Guy’s and St Thomas’ NHS Foundation Trust, Westminster Bridge Road, London SE1 7EH, UK

**Keywords:** gynaecological oncology, cytoreductive surgery, bowel surgery

## Abstract

Cytoreductive surgery for advanced ovarian cancer commonly involves bowel resection. Although UK gynaecological oncologists are trained in bowel surgery, the degree to which they perform bowel surgery independently varies nationally. A recent joint policy statement from the British Gynaecological Cancer Society (BGCS) emphasises the need for formalised colorectal support. An anonymous, online survey was emailed to BGCS members to assess the status of multidisciplinary working between UK gynaecological oncology and colorectal/general surgical teams. A total of 46 members responded (8.2% response rate). There was a large variety in the involvement of colorectal/general surgical teams in preoperative planning. A total of 13% of respondents had no formalised agreement for intraoperative support, 72.1% of respondents independently performed rectal peritoneal stripping and 60.5% independently performed small bowel resection. This was reduced to 27.9% for right hemicolectomy with primary anastomosis and 16.3% for left hemicolectomy with primary anastomosis. Respondents often involved colorectal support for post-operative complications. The majority of UK gynaecological oncologists involve colorectal/general surgical teams in bowel procedures, more commonly for large bowel procedures compared to small bowel and for left colon compared to right colon procedures. A total of 16.3% of respondents independently performed all surveyed bowel procedures. Future research should examine training and experience within these groups to address this disparity.

## 1. Introduction

Gynaecological Oncology was first defined as a subspeciality of obstetrics and gynaecology in 1969 by the American Board of Obstetrics and Gynaecology, leading to the development of a certified post-residency training programme [1].

In 1982, the UK Royal College of Obstetricians and Gynaecologists (RCOG) also recognised Gynaecological Oncology as a subspecialty and highlighted the need for links between the subspecialty and general and urological surgeons [2].

Since then, the concept of a multidisciplinary approach to gynaecological oncology surgery has become essential. Systematic review demonstrates patients with advanced ovarian cancer benefit from treatment in dedicated gynaecology oncological centres, with subspecialist and multidisciplinary input [3,4].

Surgical treatment of advanced ovarian cancer centres around cytoreductive surgery, which requires appropriate training in radical pelvic surgery, including colorectal and urological procedures. However, surgical practice varies internationally, as do survival outcomes [5]. Comparison of 5-year survival for ovarian cancer has shown that the UK falls behind other European countries. Between 2000 to 2007, this was at 31.0%, compared to 41.1% in northern Europe [6].

To standardise surgical training, the European Society of Gynaecological Oncology (ESGO) has set out a curriculum for trainees [7]. Furthermore, the 2022 joint policy statement from ACPGBI, ASGBI, AUGIS and BGCS outlines governance frameworks for maximal effort cytoreductive surgery, including multidisciplinary working between gynaecological oncologists and colorectal and UGI surgeons [8]. These two frameworks highlight the important balance between multidisciplinary support from other surgical colleagues, and appropriate training for gynaecological oncology trainees to achieve proficiency in surgical procedures.

The policy statement highlights the importance of involving colorectal surgeons prior to surgery if their input is anticipated. The statement also recognises that the degree to which colorectal surgeons will be involved in bowel surgery will vary depending on the experience of the gynaecological oncologist and surgical centre. Nevertheless, protocols for the postoperative management of these patients should be agreed between both gynaecological oncology and colorectal/UGI surgical departments. The statement consequently highlights the importance of appropriate colorectal training for gynaecological oncologists.

This study consequently sets out to assess the current status of multidisciplinary work between gynaecological oncology and colorectal/general surgical teams in the UK.

## 2. Materials and Methods

An online survey was emailed to all 569 members of the British Gynaecological Cancer Society (BGCS) in December 2021 using SurveyMonkey, with a reminder email sent four weeks later. A preliminary scoping survey was emailed one year prior, with 22 responses, helping to determine which themes to explore in more depth.

Survey data were collected anonymously, with 28 multiple-response questions and opportunities to provide open responses. The survey was divided into two sections. The first section ascertained the working habits, training and access to colorectal surgeons in the pre- and peri-operative periods.

The second section explored the role of colorectal/general surgical support in common bowel procedures performed during cytoreductive surgery and their input with surgical complications and post-operative management. The survey questions and answer options are detailed in Appendix A.

A core outcome set (COS) was not used, as there are none in development regarding ovarian cancer at present.

Data were processed and analysed in excel, with descriptive statistics detailed below. Ethical approval was not required for this survey.

## 3. Results

A total of 46 surgeons responded (8.2% response rate) to the first section of the survey, with 43 (7.7%) of these also completing the second section.

A total of 93.5% of respondents described themselves as a consultant gynaecological oncologist, with the remaining respondents including fellows and trainees. 

A total of 87.0% of respondents were subspecialists trained in gynae-oncology with GMC accreditation. One respondent was not GMC-accredited but had completed an advanced training skills module (ATSM). A total of 10.9% were not subspecialists, again including fellows and trainees, as well as gynaecologists prior to starting subspecialty training and a medical oncologist.

A total of 95.6% of respondents worked in a tertiary centre, and 87.0% had formal training in exenterative surgery, including bowel resection with anastomosis or stoma formation.

A total of 91.3% had access to colorectal/general surgical support onsite, with the remaining 8.7% having access on a remote basis.

Ovarian cytoreduction surgery made up over 40% of operating cases for 41.3% of the surveyed respondents.

Regarding primary surgical debulking (PSD) of advanced ovarian cancer (stage 3 and 4), 54.4% of respondents commented that more than half of patients would receive PSD rather than neoadjuvant chemotherapy. 

Concerning advanced ovarian cancer surgery, 17.4% had a dedicated colorectal surgeon routinely attending cytoreductive surgery, 69.6% had colorectal support available on a standby basis when required intraoperatively, and 13.0% relied on the on-call team to attend if colorectal support was required intraoperatively, with no formalised agreement.

With regard to the initial MDT pre-operative assessment, 26.1% had formal colorectal/general surgery input, 37.0% had such colorectal/general surgery available for advice on preoperative planning, and 37% had no input from such teams in preoperative planning of cytoreductive surgery patients.

Within Section 2 of the survey, respondents answered questions on the role of colorectal/general surgical support. A total of 19.6% reported that colorectal/general surgical teams routinely attend cases where bowel resection is anticipated. When bowel surgery becomes necessary intraoperatively, 28.3% reported such teams are called to attend and undertake the bowel surgery, while 13.0% reported that these teams will assist with diagnostics and the direct supervision of bowel surgery undertaken by gynae-oncology. The remaining 39.1% reported that colorectal/general surgical teams remain remote, and gynae-oncology proceed with bowel resection independently.

Respondents answered questions on different colorectal procedures; the results are detailed in Table 1.

Respondents were most confident when independently performing mesenteric/serosal disease (74.4%), rectal peritoneal stripping (72.1%), and small bowel resection with ileostomy (60.5%) or primary anastomosis (60.5%). In contrast, respondents were less likely to undertake procedures of the large bowel. Colorectal/general surgical teams would be more likely to undertake left hemicolectomy with recto-sigmoid resection and primary anastomosis (53.5%), transverse colectomy with primary anastomosis (48.9%) right hemicolectomy with primary anastomosis (48.8%), and sigmoid colectomy with primary anastomosis and de-functioning ileostomy (48.8%). Across large bowel procedures, respondents were less likely to perform procedures independently where such procedures involved primary anastomosis, compared to stoma formation.

Table 2 details how respondents would involve colorectal surgeons in peri-operative complications. A total of 83.3% of respondents felt confident independently repairing non-full-wall thickness or serosal bowl injury, compared to 57.1% of respondents when bowel injury is at full thickness. This reduced to 42.9% where bowel resection was required, and 38.1% if de-functioning loop colostomy was required. For respondents who provided open answers, they highlighted that the decision to involve colorectal colleagues was dependent on the area of bowel that was affected and if the patient had undergone bowel preparation. Others highlighted that colorectal colleagues would provide support mainly for clinical governance or medico-legal reasons.

Table 3 summarises the input of colorectal support in post-operative management. With regard to the establishment of oral intake, 67.4% of respondents rarely seek colorectal advice or support, whereas 48.8% of respondents would always involve colorectal teams in post-operative complications such as anastomotic leak or stoma breakdown. Open responses acknowledged the role of joint ward rounds with colorectal colleagues, with others highlighting that gynae-oncology teams would diagnose complications before liaising with colorectal colleagues regarding necessary intervention.

## 4. Discussion

Main Findings:

This study demonstrates national discrepancies in colorectal/general surgical support for patients with advanced ovarian cancer. A total of 13% of respondents had no formalised agreement for intraoperative support from colorectal/general surgical teams, instead relying on on-call teams. This survey also shows that gynaecological oncology surgeons most often undertake bowel procedures with colorectal support. However, particularly for bowel resection with stoma formation or anastomosis, a large proportion of procedures are performed purely by colorectal teams.

Although nearly 90% of respondents would always or often involve colorectal support for post-operative complications, over two thirds of respondents rarely seek support for post-operative instructions such as establishment of oral intake. This may reflect standardised post-operative protocols for aspects such as establishing oral intake. In fact, open responses highlighted the role of joint postoperative ward rounds with colorectal colleagues.

Strengths and Limitations:

One weakness of this study is a low response rate. This may be due to the survey email being missed by recipients. The length of the survey, at 28 questions, may have also deterred recipients from completing it. Three of the 46 respondents only completed the first section of the survey. Furthermore, it is unclear if the survey population represents surgeons that are more or less experienced in bowel procedures.

Nevertheless, the survey length does facilitate an exploration of colorectal support in the pre-, peri- and post-operative stages of cytoreductive surgery, alongside a large range of bowel procedures commonly performed during cytoreductive surgery.

Interpretation:

The findings of this study contradict the BGCS joint policy statement that colorectal colleagues should only be called on in emergency situations in exceptional circumstances [8]. Similarly, over one third of respondents had no input from colorectal/general surgical teams in the preoperative planning of cytoreductive surgery, contrasting the recommendation for colorectal surgeons to be involved prior to anticipated surgery [8].

The ESGO Gynaecology Oncology curriculum expects trainees or fellows to undertake a minimum of 20 cytoreductive surgeons as first surgeon. These should include either bowel resection, upper abdominal procedures or bulky lymph node resection [7]. The UK is one of several European countries that accepts the ESGO postgraduate training system. The findings of this survey, therefore, show that although gynaecological oncologists are expected to be able to perform such surgeries, in reality, colorectal colleagues often perform these procedures instead. The BGCS joint policy statement does acknowledge that the degree of support required from colorectal teams and their involvement in bowel procedures will vary depending on the skills of the gynaecological oncologists at different centres [8]. In 2009 data from one tertiary centre in North East England, 91% of bowel procedures in ovarian cancer patients were performed independently by gynaecological oncologists, demonstrating that this variation has existed for several years [9]. The UK Royal College of Obstetricians and Gynaecologists (RCOG) has set out a definitive document on subspecialty training in Gynaecological Oncology, outlining that trainees should assess and perform appropriate surgery on the gastrointestinal tract. The document suggests that proficiency in this outcome can be gained through attachment to a colorectal surgical team or attendance at colorectal outpatient clinics, although such experiences are not mandated [10]. In contrast, several US Gynaecological Oncology training programmes include obligatory rotational placements in colorectal and general surgery [11]. Future research should consequently examine whether such practices reflect a lack of appropriate training or preferred operating practices. Respondents did highlight that this was often for clinical governance or medico-legal purposes.

The surgical approach to bowel resection also differs for cytoreductive surgery compared to bowel resection for a primary bowel cancer. The aim of debulking is to remove visible disease to improve the effects of chemotherapy agents [12]. In comparison, colorectal surgeons aim to remove bowel cancer with a minimum 5 cm margin, removing the blood supply and lymphatics at the level of the primary feeding vessel’s origin [13]. Such an approach is unnecessary here and would lead to ovarian cancer patients having more bowel and mesentery resected than is necessary, reducing blood flow and increasing anastomotic leak risk [14]. Given the differences in operating practices, the decision to involve colorectal/general surgical teams should ultimately be made at the discretion of the gynaecological oncologist. However, the results of our survey suggest this is not the case.

In a 2018 survey of Spanish trainees in the European Network of Young Gynaecological Oncology (ENGYO), 78.4% of respondents identified bowel surgery as an area of training they felt was most required [15]. For small bowel resection, 13.5% of respondents felt comfortable operating as the first surgeon. For colorectal resection, this was 2.7%. Although this survey focused on trainees rather than consultants, it does suggest a lack of appropriate training. The greater preparedness to undertake small bowel procedures compared to large bowel procedures is similar to the findings in this study.

Our findings may also relate to opinions on cytoreductive surgery within gynaecology oncology in the UK. A survey of gynaecological oncologists in the UK, Norway, Australia, Denmark and Canada showed a significant difference in their opinions on ‘ultra-radical’ surgery for patients with advanced ovarian cancer [5]. All Australian and Norwegian respondents agreed or strongly agreed with such surgery. In Canada and the UK, this was less apparent, with some respondents disagreeing with this approach. Interestingly, willingness to undertake such surgery was found to correlate with 3-year survival [16].

The BGCS policy statement does recommend that post-operative management follows locally agreed protocols, with the management of complications agreed between gynaecological oncology and colorectal/UGI teams [8]. This survey, therefore, suggests that this is one aspect of the policy statement that is being achieved.

## 5. Conclusions

Colorectal and general surgical colleagues are commonly involved in managing post-operative complications of advanced ovarian cancer. However, there is variation in the involvement of these teams in pre-operative planning. Despite nearly all survey respondents working in tertiary care centres, on-site colorectal support is not always available and, in some instances, there is no formal agreement regarding intraoperative colorectal support. The findings of this study, therefore, suggest that the governance frameworks set out in BGCS’ joint policy statement are not being fully implemented. While some centres perform bowel procedures almost completely independently, others are much more reliant on colorectal or general surgical teams.

For centres where gynaecological oncologists are performing most bowel procedures independently, future research should examine useful lessons that can be implemented elsewhere, for example, more training in large bowel surgery. It would be useful to conduct qualitative research to build on this study. Through semi-structured interviews, it may be easier to explore the barriers that exist to multidisciplinary working within advanced ovarian cancer and suggest potential solutions to addressing the variation in colorectal/general surgical support.

As the joint policy statement suggests a reciprocal relationship regarding gynaecological support in colorectal procedures, it would also be useful to survey colorectal consultants and ascertain how gynaecological colleagues support them in the pre-, peri- and post-operative periods. There is a well-established national bowel cancer audit; however, no similar system exists in gynaecological oncology [17]. The joint policy statement concludes by stating the need for regular mortality and morbidity meetings and feeding data into national registries. We recommend the establishment of a separate national audit for ovarian cancer, to ensure that best practice is incentivised.

## Figures and Tables

**Table 1 jcm-11-06233-t001:** How do you perform bowel procedures with regard to colorectal/general surgical support?

	Independently with Remote Emergency Colorectal/General Surgical Support if Required	Independently with Remote Pre Planned Colorectal/General Surgical Support if Required	Colorectal/General Surgical Attend Theatre for the Purpose of Direct Supervision	Colorectal/General Surgical Would Undertake the Procedure
**Procedure**				
Small bowel resection with ileostomy	60.5%	13.4%	4.7%	20.9%
Small bowel resection with primary anastomosis	60.5%	16.3%	7.0%	16.3%
Right hemicolectomy with no anastomosis (bowel stoma)	32.6%	16.3%	11.6%	39.5%
Right hemicolectomy with primary anastomosis	27.9%	11.6%	11.6%	48.8%
Transverse colectomy with colostomy	37.2%	11.6%	4.7%	46.5%
Transverse colectomy with primary anastomosis	25.6%	11.6%	14.0%	48.9%
Left hemicolectomy with colostomy	39.5%	14.0%	11.6%	34.9%
Left hemicolectomy including recto-sigmoid resection with primary anastomosis	16.3%	9.3%	20.9%	53.5%
Sigmoid colectomy with primary anastomosis and de-functioning ileostomy	23.3%	9.3%	18.6%	48.8%
Hartmann’s procedure	46.5%	14.0%	7.0%	32.6%
Peritoneal stripping of the rectum	72.1%	16.3%	4.7%	7.0%
Resection of bowel mesenteric or serosal disease	74.4%	18.6%	4.7%	2.3%

**Table 2 jcm-11-06233-t002:** Would you undertake or diagnose the following intraoperative complications?

	No, I would Wish Colorectal to Attend from the Point of Suspected Injury	I would Feel Confident to Diagnose but Would Wish Colorectal to Attend	I would Feel Confident to Repair but Would Wish Colorectal to Attend	I would Feel Confident to Undertake Repair Independently without the Involvement of Colorectal Surgeons	Other
**Complication**					
Non full-wall thickness or Serosal bowel injury	0.0%	4.8%	11.9%	83.3%	0.0%
Full wall thickness, including mucosa, bowel injury	4.8%	4.8%	26.2%	57.1%	7.1%
De-functioning loop colostomy when performed with large bowel repair	21.4%	14.3%	23.8%	38.1%	2.4%
Bowel resection	19.1%	11.9%	16.7%	42.9%	9.5%

**Table 3 jcm-11-06233-t003:** What is the role of colorectal/general surgical teams in the post-operative period?

	Always Involved	Often Seek Colorectal Support or Advice	Rarely Seek Colorectal Support or Advice	Other
Post-operative instructions such as establishment of oral intake	9.3%	16.3%	67.4%	7.0%
Post operative complications such as anastomotic leak or stoma breakdown	48.8%	39.5%	9.3%	2.3%

## Data Availability

The data that support the findings of this study are available from the corresponding author upon reasonable request.

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
