# Peer review of "Gynae-Oncology Surgeons’ Preparedness to Undertake Colorectal Procedures during Cytoreductive Surgery for Ovarian Cancer: A Cross Sectional Survey"

_jcm, 2022, doi:10.3390/jcm11216233_

Round 1
Reviewer 1 Report
I do not have major issues with this article and the data clearly support the conclusions. It is an important article highlighting the synergy that should exist between gynae-oncologist and colorectal surgeons. I would suggest a mild addition to the article on the proposed training of gynae-oncologist that are needed for them to perform small bowel resection safely. I understand this information may not be readily available from the responses from those surveyed but would be possible to add this to the discussion based on the available training programmes for gynae-oncologists worldwide.
An excellent study and congratulations to the authors for addressing this issue
Author Response
Thank you for your kind feedback on our study. In light of your comments we will add a reference to the RCOG definitive document which outlines expectations for gynaecological oncology trainees in the UK, alongside how this contrasts training in countries such as the US.
Our addition:
The UK Royal College of Obstetricians and Gynaecologists (RCOG) has set out a definitive document on subspecialty training in Gynaecological Oncology, outlining trainees should assess and perform appropriate surgery on the gastrointestinal tract. The document suggests that proficiency in this outcome can be gained through attachment to a colorectal surgical team or attendance at colorectal outpatient clinics, although such experiences are not mandated. Our research suggests more experience and exposure to colorectal surgery is needed in UK subspecialty training. Several US gynaecological oncology training programmes include rotational placements in colorectal and general surgery.
Reviewer 2 Report
I read this original research with great interest about one of the most demanding challenges in gynecologic oncology: ovarian cancer and surgical treatment.
The multidisciplinary approach is the standard of care in gynecologic oncology, especially in ovarian cancer management.
The surgical procedure could be different and advanced during primary/interval or secondary cytoreduction. In each center there is a different collaboration between gynecological oncological surgeon and general surgeons.
The purpose of this survey is very interesting, and the survey was well conducted.
The questionary was very comprehensive and specific to better analyze and construe the results.
As the Authors defined in the manuscript, the huge limitation of the study is the very small sample and the very low surgeon’s compliance.
Author Response
Thank you for your comments. As you have highlighted, the sample size is a limiting factor and one consequence of the depth of the survey and scope of questions asked.
Nevertheless, this is an interesting finding in itself as it reflects either a lack of interest or a wish to stay away from this issue as it is a sensitive subject. In our discussion we highlight a similar survey of Spanish Gynae-Oncology trainees. This survey attracted a higher response rate (58%), albeit a smaller sample size than our study. The higher response rate suggests that those in other European countries may feel more open to speaking about this topic, particularly those still in training.
Reviewer 3 Report
Thank you for the opportunity to review this manuscript. The research idea is quite interesting and the paper is well written, unfortunately the execution needs improvement. The main issue with this paper is the very low questionnaire response rate that could have biased the results significantly. A response rate of 7.7% does not allow to draw any conclusions about the general situation with colorectal procedures within the Gynae-Oncology Surgeons community.
Author Response
Thank you for your feedback and comments on your study. We would disagree that the response rate precludes any potential conclusions about the situation within Gynae-Oncology. The response is an interesting finding in itself as it reflects either a lack of interest or a wish to stay away from this issue as it is a sensitive subject. In our discussion we highlight a similar survey of Spanish Gynae-Oncology trainees. This survey attracted a higher response rate (58%), albeit with a smaller sample size than our study. The higher response rate suggests that those in other European countries may feel more open to speaking about this topic, particularly those still in training.
The questionnaire reflects the inclination towards bowel procedures within 7.7 % of the UK community, with this group more likely to be doing such procedures within their practice. Although a definitive conclusion on the whole UK Gynae-Oncology community cannot be drawn, our survey shows that discrepancies certainly exist within the community. If the response rate was increased by the inclusion of Gynae-Oncologists less interested or experienced in bowel procedures, it is likely these discrepancies would only be further amplified.
Reviewer 4 Report
Thank you for the opportunity to review your manuscript on the preparedness of Gyn Onc surgeons to proceed with cytoreductive surgery for Ovarian Cancer. This manuscript is a clear presentation of a survey and the survey results. I have the following comments:
1. Was a pilot test of the survey completed? Was there anything in the pilot test that could have predicted the low response rate? Were any coarse adjustments made to optimize response rate?
2. In retrospect are there any adjustments that could have improved response rate aside from the length of the survey?
3. Gyn Oncs usually train in there 20's or early 30's. They practice until 60-65yo. Training and expectations for surgical care change over that 30-40year career. Not sure if you can see an age or years from training trend in your data.
Author Response
Thank you for your comments, to address them as you have asked them:
- A pilot study was sent out one year prior as a scoping exercise, with 22 responses, predicting further surveys may also receive a low response rate. This suggests either a lack of time amongst specialists to complete the survey, a lack of interest or a wish to stay away from this issue as it is a sensitive subject. The findings of our scoping survey allowed us to decide on which aspects to explore in more depth within the final survey. On reflection, it would have been useful to consider how the survey could be optimised to improve the response rate.
- Aside from the length of the survey, the method of publicising the survey could have been adjusted to improve response rate. Our method of using the BGCS email to target Gynaecological Oncologists ensured respondents were the appropriate population for the study, but the main drawback being responses to email bulletins and surveys is classically low almost universally. By attracting more centres to collaborate on the study, we may have been able to encourage more responses via word of mouth or networks, but with the chance of introducing regional biases.
- Our study did not ask respondents to provide age but we agree there likely exists clear differences between age groups. A similar study of Spanish Gynae-Oncology trainees we highlighted in our discussion attracted a higher response rate (58%), suggesting this is a topic that younger groups may have more interest in.